# The Effect of the ZrO_2_ Loading in SiO_2_@ZrO_2_-CaO Catalysts for Transesterification Reaction

**DOI:** 10.3390/ma13010221

**Published:** 2020-01-04

**Authors:** Daniela Salinas, Sichem Guerrero, Cristian H. Campos, Tatiana M. Bustamante, Gina Pecchi

**Affiliations:** 1Departamento de Química, Universidad del Biobío, Avenida Collao 1202, Concepción 4030000, Chile; 2Facultad de Ingeniería y Ciencias Aplicadas, Universidad de Los Andes, Monseñor Álvaro del Portillo 12455, Las Condes, Santiago 7550000, Chile; sguerrero@uandes.cl; 3Departamento de Físico-Química, Facultad de Ciencias Químicas, Universidad de Concepción, Edmundo Larenas 129, Concepción 4030000, Chile; ccampos@udec.cl (C.H.C.); tatibustamante@udec.cl (T.M.B.); gpecchi@udec.cl (G.P.); 4Millenium Nuclei on Catalytic Processes towards Sustainable Chemistry (CSC), Santiago 8940000, Chile

**Keywords:** core@shell, catalyst, ZrO_2_, CaO, FAME

## Abstract

The effect of the ZrO_2_ loading was studied on spherical SiO_2_@ZrO_2_-CaO structures synthetized by a simple route that combines the Stöber and sol-gel methods. The texture of these materials was determined using S_BET_ by N_2_ adsorption, where the increment in SiO_2_ spheres’ surface areas was reached with the incorporation of ZrO_2_. Combined the characterization techniques of using different alcoholic dissolutions of zirconium (VI) butoxide 0.04 M, 0.06 M, and 0.08 M, we obtained SiO_2_@ZrO_2_ materials with 5.7, 20.2, and 25.2 wt % of Zr. Transmission electron microscopy (TEM) analysis also uncovered the shape and reproducibility of the SiO_2_ spheres. The presence of Zr and Ca in the core–shell was also determined by TEM. X-ray diffraction (XRD) profiles showed that the c-ZrO_2_ phase changed in to m-ZrO_2_ by incorporating calcium, which was confirmed by Raman spectroscopy. The purity of the SiO_2_ spheres, as well as the presence of Zr and Ca in the core–shell, was assessed by the Fourier transform infrared (FTIR) method. CO_2_ temperature programmed desorption (TPD-CO_2_) measurements confirmed the increment in the amount of the basic sites and strength of these basic sites due to calcium incorporation. The catalyst reuse in FAME production from canola oil transesterification allowed confirmation that these calcium core@shell catalysts turn out to be actives and stables for this reaction.

## 1. Introduction

An efficient strategy to control carbon dioxide, CO_2_ emissions relies on developing platforms in order to reduce the consumption of fossil fuels around the world [1,2,3]. As an alternative to this type of fuel, biofuels are becoming an important source of fuels contributing to the reduction of CO_2_ emissions [3,4]. Biofuels are categorized in primary and secondary groups. The first group considers the production of biofuels from plants, animal waste, and wooded places [5]. The second group is related to the production of biofuels from raw materials and microorganisms. This group is further divided into first, second, and third generations. The first generation is produced from edible oils, such as corn, sunflower [1,6], cotton [7], soybean [8], palm [2], rapeseed [9], and canola oils [10,11]. On the other hand, second generation biofuels are from non-edible oils, such as jatropha [12,13,14], karanja [15], castor [16], and linseed [17]. The third generation of biofuels is produced from microalgae species [18,19,20,21,22]. Biodiesel represents a green alternative for diesel engines due to the renewable, clean, biodegradable, non-toxic, and environmental friendliness properties of this type of fuel [5,10,17,23]. Biodiesel is defined as chains of fatty acid methyl ester (FAME), which can be used as an alternative to diesel engine fuel. This green fuel is frequently obtained from the transesterification of triglycerides with methanol [3,7,17]. Traditionally, biodiesel is generated by transesterification reactions of vegetable oils using a homogenous catalyst [24,25]. Due to their large catalytic activity, homogeneous basic catalysts, such as KOH, CH_3_OK, NaOH, and CH_3_Ona, are often used in the industrial production of biodiesel [25]. While the homogeneous catalysts are widely used in the biodiesel industry, the separation of catalysts from products, the corrosion of the reactor, as well as secondary reactions, have led us to explore heterogeneous catalysts as a potential alternative to improve biodiesel production. Solid catalysts, such as MgO [2] Li/TiO_2_ [10], La_2_O_3_-ZrO_2_ [11], CaO-La_2_O_3_ [26], CaO [27,28,29], potassium-impregnated Fe_3_O_4_-CeO_2_ [9], Li_2_SiO_3_ [30], and Ag/CeO_2_ [31], have been efficiently employed for biodiesel production. Nanomaterial catalysts have gained attention in academia and industry due to their properties in tuning chemical reactions [9,32,33,34]. The main advantage of these materials is primarily attributed to the novel properties of nanoparticles comprising their nanoscale size, structure, morphology, and reactivity. In addition, nanoparticles show important features, including a high degree of crystallinity, remarkable stability, and catalytic activity. Besides this, nanoparticles can be used in food, agricultural, cosmetic, electronic, and pharmaceutical applications [5]. Core–shell materials decorated with metallic systems play an important role in heterogeneous catalysis since the transfer of electrons demonstrates advantages in the stabilization of phases, preventing the migration and sinterization of the active sites and favoring higher activity of the final catalysts [5,35]. Recently, zirconia (ZrO_2_) has received attention due to its amphoteric nature [36] and the ability to tune its chemical properties [37], electronic properties, thermal stability [38,39], and chemically inert inorganic oxide and redox properties [40]. Nonetheless, the deposition of one oxide on another oxide is interesting to understand. In general, ZrO_2_ can be partially stabilized when doped with CaO [36,41,42]. In addition, when ZrO_2_ can be partially stabilized for any doped agent, the crystalline structures can undergo changes because the ZrO_2_ has an ability to embed foreign cations in their network [11,43]. Hence, calcium has been considered as a foreign cation that facilitates basic properties in a core–shell material capable of FAME production [44]. In relation to this, core–shell catalysts have been recently been covered in a full review about biofuels [5]. In this study, the effect of ZrO_2_ loading in SiO_2_@ZrO_2_-CaO core–shell catalysts for the production of FAME from canola oil transesterification is proposed.

## 2. Materials and Methods 

### 2.1. Synthesis of SiO_2_ Spheres

The SiO_2_ spheres were synthetized by the Stöber method using tetraethyl orthosilicate (TEOS) as a silica source [45]. In a typical procedure, a certain amount of the TEOS was dissolved in absolute ethanol and mixed with ammonia solution (25 wt %), water for chromatographic Merck, LC-MS grade and absolute ethanol. Subsequently, the dissolution was stirred for 6 h at room temperature, and the colloidal suspension was centrifuged and washed several times with ethanol. The resulting material was re-dispersed in ethanol under sonication.

#### 2.1.1. Synthesis of SiO_2_@ZrO_2_

The SiO_2_ sphere suspension was dispersed in absolute ethanol and ammonia dissolution (28 wt %) under sonication. Then, alcoholic dissolutions of zirconium (VI) butoxide 80 wt % in butanol solution Sigma-Aldrich (0.04 M, 0.06 M, and 0.08 M) were added to the SiO_2_ suspension under stirring at room temperature [46]. After that, this suspension was stirred at 85 °C under refluxing conditions for 3 h. The silica–zirconium material was collected by centrifugation, washed several times in ethanol, dried at 70 °C for 12 h, and subsequently calcined at 700 °C in air for 6 h.

#### 2.1.2. Synthesis of SiO_2_@ZrO_2_-CaO

The surface of the SiO_2_@ZrO_2_ core–shell material was modified by solvo-thermal synthesis [47]. The SiO_2_@ZrO_2_ structures were stirred with ethanol absolute reagent for 15 min, and this mixture was named dissolution 1. C_6_H_12_O_6_∙H_2_O (2.00 g, Merck) and H_2_NCONH_2_ (0.440 g, Merck) were dissolved in HPLC water grade Merck (20 mL), and 0.250g of Ca(NO_3_)_2_·4H_2_O Sigma-Aldrich precursor was incorporated over the solution. The resulting mixture was kept for 15 min under stirring, and this mixture was named dissolution 2. After that, dissolution 2 was added dropwise into dissolution 1 under stirring for an additional 15 min. The formed nanoparticles were transferred into a Teflon-lined stainless-steel autoclave Parr reactor and maintained for 20 h at 160 °C. The material was washed several times in ethanol absolute and dried at 70 °C for 12 h, and then the solid was calcined at 700 °C in air for 6 h.

### 2.2. Catalytic Tests and Product Analysis

In the transesterification reaction (see Scheme 1), canola oil (5.00 g) was added to a three-neck round bottom batch reactor equipped with a reflux condenser, a magnetic stirred and a thermometer. As solvent, 6.02 g of anhydrous methanol 99.8% Sigma-Aldrich (molar ratio of methanol:oil = 36:1) was used at 65 °C, ambient pressure, and continuous stirring (300 rpm). A total of 300 mg of the core–shell catalyst was introduced into the reactor, corresponding to 6 wt % of the starting canola oil. The methanol and the SiO_2_@ZrO_2_-CaO catalysts were kept under stirring, and commercial canola oil was then added. The reaction time was 4 h. The evolution of the transesterification reaction was analyzed by GC-MS using the following procedure: after 4 h of reaction time and at room temperature, the products were centrifuged, and organic and aqueous phases were obtained. A total of 500 μL of the methyl esters contained in the organic phase was incorporated, with 1600 μL of the 5000 ppm icosane (C_20_H_42_, Sigma-Aldrich)/hexane solution as an external standard added to the same vial. A total of 1 µL of this mixture was injected in GC-MS. The FAME production has a relationship with commercial canola oil composition, and we considered the principal fatty acid compositions of the canola oil for catalytic activity in biodiesel production (see Appendix A). The FAME production was analyzed by a GC-MS Perkin Elmer Clarus 690-Mass Spectrometry (Claris SQ8T) using an Elite 1701 (30 m × 0.25 mm ID × 0.5 μm DF) capillary column. The programmed GC oven temperature started at 60 °C by 1 min, which then increased to 260 °C at 4 °C/min. The injection port was kept at 150 °C, using 1 ml/min of carrier helium gas and a split of 50 ml/min. The Clarus SQ8T was operated with the following parameters: SCAN mode (*m/z* = 10–300 amu); electron energy at 70 eV; transfer line 200 °C; ion source at 150 °C; and the quadrupole mass detector was in electron impact ionization mode. Chromatographic data were processed using Turbo Mass (v6.1.2.2048) and mass spectra laboratory databases (NIST 2017 v2.3).

### 2.3. Catalyst Recovery

To appraise the heterogeneity and recyclability of the SiO_2_@ZrO_2_-CaO catalysts, the solid residue in the mixture reaction was recovered by centrifugation and washed with ethanol absolute five times. After the transesterification reaction was completed (4 h), the solid was dried at 70 °C for 12 h.

### 2.4. Core–Shell Characterization

The morphology of the SiO_2_ spheres were evaluated by transmission electron microscopy (TEM, JEOL model JEM-1200 EX II microscope) and scanning electron microscopy (SEM, JEOL model JSM-6380-LV). Chemical composition was investigated by energy dispersive spectroscopy (EDS) line scanning using (SEM, JEOL model JSM-6380-LV). The textural properties were measured by N_2_ physisorption using a Micromeritics Tristar II 3020 surface area and porosity analyzer. The samples were degassed at 300 °C for 2 h prior to the measurement. The specific surface area (S_BET_) was calculated according to the Brunauer–Emmett–Teller (BET) surface areas of the core–shell, which were acquired from nitrogen adsorption isotherms at a relative pressure of 0.99. The FTIR spectra were recorded between 4000 cm ^–1^ and 500 cm ^–1^. The infrared (IR) measurements were acquired using a FTIR Nicolet Nexus, and the spectra were recorded with a 4 cm^–1^ resolution and 250 scans. The crystallinity of the sample was analyzed by the powder x-ray diffraction (XRD) spectra were obtained using a D4 Endeavor Bruker AXS diffractometer equipped with nickel-filtered Cu K α1 radiation ( λ = 1.542 Å. The standard scan parameters were 1° min^−1^ for a 2θ range of 10°–90°. Identification of the phases was carried out by reference to EVA diffraction file data. Raman spectra were acquired at room temperature with a confocal XploRA™ Plus spectrometer (Horiba Scientific), using a 532 nm diode laser (15 mW laser power) and an X50 air objective. The confocal pinhole was set to 100μm and the Raman signal was acquired using a 600 lines/mm grating centered between 100 and 5000 cm^−1^. The integration time was 50 s average time for all measurements. The temperature-programmed desorption of CO_2_ (CO_2_-TPD) was carried out in a fixed-bed reactor using 50 mg of sample diluted in sand (50–70 mesh particle size, Sigma-Aldrich). This mixture was calcined in a flow of 10 ml/min of pure oxygen using a ramp of 10 °C/min up to 650 °C and then cooled in ultra-pure He until reaching room temperature. Afterwards, 30 ml/min of 5%CO_2_/He was flown for 1 h followed by purging with helium for another h. Then, the temperature was ramp up to 650 °C at 10 °C/min in the presence of He, and the reactor effluent gases were monitored on-line by a quadrupole mass spectrometer (Omnistar, Pfeiffer Vacuum) equipped with a secondary electron multiplier (SEM) and Faraday detectors. 

## 3. Results

### 3.1. Reproducibility and Thermal Stability of the SiO_2_ Sphere

In order to define the morphology and thermal stability of the synthesized materials, SEM-EDS and TEM analysis were carried out for two calcination temperatures: 500 °C and 700 °C. In Figure 1a, we show the regular shape of the SiO_2_ spheres and its thermal stability at 700 °C. Due to this, the shape of the sphere was conserved at this calcination temperature and the purity is shown by the EDS analysis in Figure 1b. Hence, the route of synthesis and reproducibility was reached for the synthesis of the SiO_2_ spheres. Besides this, the TEM images for SiO_2_ fresh and calcined spheres are incorporated in Appendix A. The first one shows the SiO_2_ fresh sphere (without calcination) that was synthetized uniformly. Appendix A shown the SiO_2_ spheres calcined at 500 °C, and the last micrograph displays the SiO_2_ spheres calcined at 700 °C (Appendix A). The shape of the spheres was conserved, and the calcination tempearure seems not to be a relevant parameter for this study.

The calcination temperature for the SiO_2_@ZrO_2_ material at 500 °C can be seen in Appendix A. The XRD profiles showed poor crystallinity for the calcined silica spheres, and for SiO_2_@ZrO_2_, this was (2) 0.04 M and (3) 0.06 M of zirconium (IV) butoxide precursor. For this reason, the ZrO_2_ was not able to form when the materials were calcined at 500 °C. However, in SiO_2_@ZrO_2_ with (4) 0.08 M of zirconium (IV) butoxide precursor, the crystalographic phases for ZrO_2_ appeared in the core–shell.

With regard to setting the calcination temperature, the XRD profiles at 500 °C and 700 °C were essential for defining the best system and for continuing this core–shell study. Although TEM analysis not was a decisive technique, because the shape of the spheres was conserved at 500 °C and 700 °C, the XRD profile was a important characterization when set to 700 °C; thus, all materials showed ZrO_2_ crystallinity.

### 3.2. Textural Characterization for Core–Shell

#### 3.2.1. BET Surface Area

The BET surface area and the distribution of the pore size of the SiO_2_ spheres, SiO_2_@ZrO_2_ and SiO_2_@ZrO_2_-CaO structures were estimated by the physical adsorption of N_2_. The N_2_ adsorption–desorption isotherms of the SiO_2_@ZrO_2_ core–shell calcined at 700 °C for different concentrations of zirconium precursor are shown in Figure 2.

In Figure 2, a type IV isotherm can be seen, suggesting a typical mesoporous texture in agreement with IUPAC’s classification [48], and the pore size distribution peak at around 4.0 nm indicates the presence of mesopores. The S_BET_ values of the fresh spheres and non-calcined SiO_2_@ZrO_2_ materials shown in Appendix A indicate that the presence of ZrO_2_ increases the surface area in SiO_2_ spheres from 17 up to 150 m^2^g^−1^ for SiO_2_@ZrO_2_ with a decrease after the calcination temperature to 45, 23, and 36 m^2^g^−1^, respectively (see Table 1). Regarding this, an obstruction of the pores in the core–shell should be occurring. When calcium was incorporated in core–shell, the surface area decreased below 20 m^2^g^−1^ and the pore diameter increased up to 6 nm, indicative of a special interaction between Zr and Ca. The N_2_ adsorption–desorption isotherms for SiO_2_@ZrO_2_–CaO are attached in the Appendix A.

#### 3.2.2. TEM Analysis

The morphology of the synthesized samples was determined by TEM analysis. Figure 3 shows the TEM micrographs with large amount of poor dispersed and aggregated particles for SiO_2_@ZrO_2_ 0.04 M (Figure 3a). Meanwhile, with the increased of the zirconium oxide loading, the ZrO_2_ coverage is more homogeneous according to Figure 3b,c, respectively. Therefore, it can be seen that an appropriate coverage was achieved for 0.08 M concentration of the zirconia precursor, as the best recovery was observed by TEM.

Figure 4 shows the micrographs of the SiO_2_@ZrO_2_-CaO calcined at 700 °C with the addition of the same amount of calcium precursor. Well-defined spherical structures with a ring of light material can be clearly seen in Figure 4, which were not seen in the SiO_2_@ZrO_2_ materials; this was attributed to the incorporation of calcium oxide. Moreover, the SiO_2_@ZrO_2_—0.06 M (Figure 4b,c) and SiO_2_@ZrO_2_—0.08 M (Figure 4d) materials showed a better incorporation of CaO, attributed to the high affinity between zirconium oxide and calcium oxides. 

### 3.3. SEM-EDS Analyses

SEM-EDS analyses were performed to explore the morphology and also to confirm the Zr and Ca elements, as shown in Figure 5; the EDS analysis confirms the presence of zirconium oxide in relation with the zirconium precursor amount. The EDS analysis displayed 5.7, 20.2, and 25.2 wt % of Zr for SiO_2_@ZrO_2_ for the 0.04 M, 0.06 M, and 0.08 M, respectively. Therefore, it can be concluded the incorporation of calcium on SiO_2_@ZrO_2_ by EDS analysis from SEM.

Figure 6 shows the SEM micrograph and EDS analysis for SiO_2_@ZrO_2_ 0.06 M–CaO. The EDS spectrum for the core–shell displayed 3.1 wt % Ca for SiO_2_@ZrO_2_ 0.06 M–CaO and 5.3 wt % Ca for SiO_2_@ZrO_2_ 0.08 M–CaO. For the SiO_2_@ZrO_2_ 0.04 M–CaO sample, the wt % Ca was below 1% (not shown).

### 3.4. FTIR Analysis

The FTIR technique allows the identification of the chemical bond formation of Si-O-Zr in the core–shell, as well as the Ca-O bond interaction. Figure 7a shows the silanol Si-OH bond in SiO_2_ spheres at around 500 cm^−1^ and the asymmetric stretching vibration for Si-O-Si at 1100 cm^−1^ for SiO_2_ spheres [49]. The transmittance bands at 948 cm^−1^ and 800 cm^−1^ are associated with Si-OH bending vibration and Si-O bending vibrations, respectively. The band around 1637 cm^−1^ corresponds to Si-H_2_O flexion and the FTIR band at 3415 cm^−1^ to Si-OH stretching mode vibration (Figure 7a). When zirconium was fully incorporated to the SiO_2_ spheres, it was expected that the silanol groups of the SiO_2_ spheres should be modified [46]. Therefore, the decrease of the Si-O-Si bond intensity in Figure 7 is attributed to their disappearance considering the formation of new Si-O-Zr bonds. The band at 1100 cm^−1^ undergoes a shift at 1097 cm^−1^ caused by the Si-O-Zr formation, and the degree of the band shift depends on the number of Si-O-Zr bonds, in line with the increment in zirconium precursor concentration [46]. Therefore, because of the surface cover by ZrO_2_, it is also expected that there would be a decrease in the intensity of FTIR bands for all modes of vibration of the Si-OH and Si-H_2_O flexion, as well as the appearance of the Zr-O bonds. The zoom spectra between 400 cm^−1^ and 950 cm^−1^ shown in Figure 7b indicate the appearance of absorption bands around 556 cm^−1^ and 820 cm^−1^ associated with a Zr-O stretching vibration signal present in SiO_2_@ZrO_2_ 0.04 M, SiO_2_@ZrO_2_ 0.06 M, and SiO_2_@ZrO_2_ 0.08 M core–shells, respectively [50].

Figure 8 shows the FTIR for SiO_2_@ZrO_2_–CaO calcined at 700 °C. The FTIR indicates the typical absorption bands of CaO recorded within a range of 400–4000 cm^−1^ and reveals the band of the Ca-O bond at around 407 cm^−1^ and 707 cm^−1^ for the extra stretching and stretching vibrations, respectively. The bands at 1490 cm^−1^ and 809 cm^−1^ were attributed to the asymmetric stretching of C=O and attributed to the absorbed CO_2_ (see Appendix A, where a thermogravimetric analysis (TGA) of CaNO_3_·4H_2_O was achieved) [51]. The displayed bands of 1118 cm^−1^ for Si-O-Zr in the core–shell and 3428 cm^−1^ shows the presence of the –OH group stretching, considering the physically adsorbed water on the surface of zirconium [34,52]. 

### 3.5. X-Ray Powder Diffraction

The crystalline phases and composition of the SiO_2_@ZrO_2_ and SiO_2_@ZrO_2_-CaO core–shell calcined at 700 °C was confirmed by the XRD technique, and the respective XRD at 500 °C (Appendix A) is shown in the Appendix A. Figure 9a shows the XRD patterns for SiO_2_ spheres and the core–shell, SiO_2_@ZrO_2_–0.04 M, SiO_2_@ZrO_2_–0.06 M, and SiO_2_@ZrO_2_–0.08 M, respectively. The XRD shows a typical broad peak at around 20°, assigned to amorphous silica [49,53]. When SiO_2_@ZrO_2_ 0.04 M was calcined at 700 °C, the first evidence of the cubic or tetragonal zirconium oxide (c-ZrO_2_ or t-ZrO_2_) appears at 30.3°. The main peaks for the cubic phase of ZrO_2_ appeared at 30.3°, 35.3°, 50.2° and 60.2° (JCPDS 27-0997), similar to the tetragonal phase (JCPDS 80-0965) and monoclinic phases of ZrO_2_ at 24.2°, 28.2°, 31.4°, and 34.3° (JCPDS 37-1484) [54]. For SiO_2_@ZrO_2_ 0.04 M, the material with the lowest zirconium precursor content, the diffraction peaks are poorly defined and poorly crystalline; however, the associated signals could be assigned to c-ZrO_2_ and/or t-ZrO_2_ [55]. Noticeably, the crystallinity increases with zirconium content, with the identification of c-ZrO_2_ or t-ZrO_2_, and also m-ZrO_2_ was detected with a small diffraction pattern at 34.6° in SiO_2_@ZrO_2_ 0.06 M and SiO_2_@ZrO_2_ 0.08 M [54]. Even though the diffraction peaks of the X-ray profiles for cubic and tetragonal phases are similar, it was not possible to identify the presence of cubic and/or tetragonal ZrO_2_ phases by XRD.

When calcium was incorporated (see Figure 9b) the Zr and Ca interaction was clearly identified by XRD due to the signal at 30.3° decreasing in SiO_2_@ZrO_2_ 0.04 M–CaO, and the main peaks for m-ZrO_2_ obviously increased. Moreover, the signal for the m-ZrO_2_ is clearly seen in (3)-Ca and (4)-Ca and in the CaO diffraction peaks that appear at 32.43°, 37.14°, 53.65°, 64.49°, and 67.24° (JCPDS 48-1467) [56]. Unfortunately, the peak of CaO at 32.43° was not identified due to the large intensity of the diffraction peak at 30.3° for c-ZrO_2_ and/or t-ZrO_2_. Meanwhile, the detected diffraction peaks at 38°, 64°, and 66° can be related to CaO [25,44]. On the other hand, the detection of CaCO_3_ can be attributed to the uncompleted oxidation of the calcium precursor (see Appendix A) and/or carbonatation of the calcium core–shell. In Figure 9b, the diffraction peaks at 40°, 44.5°, 57°, and 58° confirmed the carbonate presence attributed to the absorbed CO_2_ [57]. Regarding this, a similar behavior was observed by FTIR analysis. The intensity of these calcium peaks is similar, in agreement with the fact that they were synthetized with the same amount of calcium precursor. Therefore, the presence of CaO has a clear effect on the crystallographic changes of zirconia, making a change in the disposition of the Zr-O covalent bonds and allowing the increase of the m-ZrO_2_ crystal phase. The XRD profiles for (2)-Ca and (3)-Ca are very similar with the presence of CaO, CaCO_3_, t-ZrO_2_, and m-ZrO_2_ crystalline phases.

### 3.6. Raman Spectroscopy

In order to confirm the zirconium and calcium interaction, Raman spectra are reported in Figure 10. For the SiO_2_ sphere, the Raman bands at 430 cm^−1^, 605 cm^−1^, and 804 cm^−1^ (see Figure 10a) are associated with silica nanoparticles [58]. In relation to Alessi et al., the Raman spectrum of the core is basically that recorded in nanoparticles with a lower specific area and is comparable with the bulk silica. Therefore, the Raman band at 430 cm^−1^ is associated with the oxygen in the Si-O-Si vibration. At 605 cm^−1^ and 800 cm^−1^, the Raman bands are also assigned for SiO_2_ nanoparticles, and their band intensity depends on the size and type of the spheres [59]. 

Basahel et al. reported Raman spectrum for zirconia and their effect in the application of photocatalysis [54]. With reference to t-ZrO_2_, seven Raman-active bands at 149, 224, 292, 324, 407, 456, and 636 cm^−1^ and m-ZrO_2_ show nine bands at 183, 301, 335, 381, 476, 536, 559, 615, and 636 cm^−1^, as reported in Figure 10a (curve 1); the Raman bands for SiO_2_@ZrO_2_ samples indicate the other crystallographic phases, and the cubic crystallographic phase (c-ZrO_2_) for core–shell samples over the tetragonal phase can be seen by the XRD profiles. Despite the similar wavenumber of the Raman bands for the tetragonal and cubic phases, the broad bands for amorphous c-ZrO_2_ show large differences. According to Basahel et.al, the c-ZrO_2_ phase is characterized by broad Raman bands at 246, 301, 436, and 625 cm^−1^, whereas the shape of the Raman spectra is related to the amorphous phase, which is due the relocated oxygen atoms in lattice sites. On the other hand, the tetragonal ZrO_2_ phase is more symmetric. Sample (2) in Figure 10a shows Raman spectra similar to the crystallographic phase of c-ZrO_2_, which then develops as the content of zirconia increases up to 0.08 M (see Figure 10a, curve (4)). The Raman spectra of the SiO_2_@ZrO_2_ –CaO samples were also obtained, and they are shown in Figure 10b. From this figure, it is observed that (2)-Ca and (3)-Ca samples present a typical Raman spectrum for c-ZrO_2_, with broad Raman peaks at 143, 270, 454, and 633 cm^−1^. Finally, when the zirconium oxide increased further (Figure 10b, curve (4)-Ca), the monoclinic phase became very clear at 360 cm^−1^ together with the appearance of the CaO signal [60].

### 3.7. Carbon Dioxide Temperature-Programmed Desorption 

The basicity of the catalysts was measured by CO_2_-TPD. This technique allows the description of the strength of the basic sites according to the CO_2_ desorption temperature, as weak (T < 200°C) and associated with –OH groups, medium strength (between 200 °C and 400 °C) associated with adsorbed bidentate carbonates, and strong basic sites (T > 400 °C) associated with monodentate carbonates formed as O^2−^ ions [11]. The CO_2_-TPD results indicated (see Appendix A for desorption profiles) that the SiO_2_@ZrO_2_ samples have very few basic sites (28, 69, and 46 μmol g^−1^ for SiO_2_@ZrO_2_ 0.04 M, SiO_2_@ZrO_2_ 0.06 M, and SiO_2_@ZrO_2_ 0.08 M, respectively). On the other hand, it was expected that the addition of calcium oxide could be able to provide basic sites to the core–shell materials. When the same amount of calcium oxide was added to the SiO_2_@ZrO_2_ 0.04 M, SiO_2_@ZrO_2_ 0.06 M, and SiO_2_@ZrO_2_ 0.08 M, strong basic sites were found in all the samples. According to the results in Table 1, the total amounts of basic sites were 273, 306, and 454 μmol g^-1^ for the SiO_2_@ZrO_2_ 0.06 M–CaO and SiO_2_@ZrO_2_ 0.08 M–CaO samples, respectively (see Appendix A for desorption profiles). In this context, the increase in ZrO_2_ content seems to favor the interaction with the CaO phase. In summary, it seems that the m-ZrO_2_ phase is associated with the strong basicity of the catalysts. 

### 3.8. Reusability SiO_2_@ZrO_2_-CaO Core–Shell

From an economic point of view, the core–shell synthesis and reusability of the catalysts are relevant issues for FAME production. The main requirements for the heterogeneous catalyst in the transesterification reaction are stability, activity, and selectivity. In this context, the SiO_2_@ZrO_2_-CaO materials were tested as heterogeneous catalysts in FAME production from canola oil via the transesterification reaction. Regarding the catalytic activity, all the catalysts were active in the tested reaction, following the trend SiO_2_@ZrO_2_ 0.08 M-CaO > SiO_2_@ZrO_2_ 0.06 M-CaO > SiO_2_@ZrO_2_ 0.04 M–CaO, which is in line with the basic properties of these materials (see Table 1), due to the CaO and m-ZrO_2_ contributions. In relation with this, the m-ZrO_2_ provides a higher concentration of basic sites and the CaO-stabilized ZrO_2_ allowed the crystallographic phase to change from cubic to monoclinic zirconia, providing an important amount of total base sites [43]. The main products result to be 9-octadecanoic acid (Z)-methyl esters, (C_19_H_36_O_2_) from C_18:1_ oleic fatty acid composition and 9,12-octadecanoic acid (Z,Z)-methyl ester (C_19_H_34_O_2_) from C_18:2_ linoleic fatty acid composition, in relation with the fatty acid composition of the canola oil reported in Appendix A [61].

After a catalytic evaluation, these catalysts were separated from the liquid phase. Each catalyst was washed with ethanol absolute five times and dried in air at 70 °C around 12 h without a new calcination procedure. In Figure 11, a slight activity decrease was observed in both SiO_2_@ZrO_2_ 0.08 M–CaO and SiO_2_@ZrO_2_ 0.06 M–CaO catalysts from the first to fourth reaction cycle, with FAME production after 4 h equaling 74% and 55%, respectively. This behavior could be attributed to the atmosphere exposure of the catalysts during the dried treatment after the catalytic measurements. According to the characterization results, both catalysts showed the presence of CaCO_3_ as an impurity, attributed to an incomplete oxidation of the calcium precursor during the material synthesis and/or to the CO_2_ adsorption from the atmosphere. We suggest that the exposure of the catalytic system with the atmosphere promotes the adsorption of CO_2_ from the atmosphere decreasing the CaO available to catalyze the transesterification reaction.

## 4. Conclusions

This study shows the calcium effect when added to the SiO_2_@ZrO_2_ core–shell. First, the calcium oxide allowed a change in crystallographic phases from cubic to monoclinic ZrO_2_ phases. Although no XRD profile was able to detect the crystallographic phases for zirconia because tetragonal and cubic phases have a similar XRD pattern, Raman spectra displayed clear differences between cubic and tetragonal phases when calcium was added because one broad spectrum is characteristic for c-ZrO_2_. In fact, the Raman spectra showed unique c-ZrO_2_ from SiO_2_@ZrO_2_ 0.04 M–CaO, allowing a confirming phase after being viewed by the XRD. Although no XRD profile allowed us to confirm the presence of c-ZrO_2_, it was a complementary technique that allowed the confirmation of the presence of crystallographic phases, while m-ZrO_2_ was obviously detected by XRD when calcium oxide is in the core–shell. The Raman spectrum for SiO_2_@ZrO_2_ 0.06 M–CaO resulted in a crystallographic transition from c-ZrO_2_ to m-ZrO_2_ phases, and SiO_2_@ZrO_2_ 0.08 M–CaO was more obvious in the core–shell of m-ZrO_2_ and CaO. For these reasons, the CaO partially stabilized zirconia. Secondly, the calcium oxide improved the total number of basic sites. Due to this, the increment in the basic sites is associated with strong basic sites for calcium oxide and the presence of monoclinic ZrO_2_. The characterizations shown in this study allowed the SiO_2_@ZrO_2_ 0.08 M–CaO to be a more active catalyst in the transesterification reaction. Thus, this core–shell showed stronger basic sites and higher amounts of the basic sites not only for the contribution of the calcium oxide but also the presence of m-ZrO_2_, with close to 80% of FAME production.

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
