# Peer review of "The Effect of the ZrO2 Loading in SiO2@ZrO2-CaO Catalysts for Transesterification Reaction"

_materials, 2020, doi:10.3390/ma13010221_

Round 1

Reviewer 1 Report

Salinas and co-workers reported the effect of the ZrO2 loading in SiO2@ZrO2-CaO2 catalysts for the transesterification reaction. They found the important role of CaO stabilizing and changing the ZrO2 phase from cubic to monoclinic zirconia generating basic sites that are important for the target reaction. However, the conclusion was very weakly supported by experiments and English needs to be greatly improved to be published. Therefore, I could not recommend to be published this paper in its current form. Following is my comments:

Overall, English needs to be greatly improved to be published in Materials. Following is a few examples:

        In line 110, “were keep” should be changed to “were kept”

        In line 178, “was not able to forming” should be changed to “was not            able to form”

        In the line 200, “bellow” should be changed to “below”

        The sentence shown in the line 253-254 is not grammatically correct.  

Please replot the EDS pattern shown in Figure 1 (b), Figure 5, and Figure 6(b) with exported raw data to look better. For example, the spectrum is only shown up to 4kev, so it doesn’t need to show up to 20kev. The authors claimed that Zr concentrations were 5.7, 20.2 and 25.2 wt% when 0.04M, 0.06M and 0.08M zirconium (VI) butoxide were used to synthesize catalysts. Why there was a noticeable increase in Zr weight loading from 0.06 to 0.08M? The authors obtained the conclusion that the basic properties of these materials are an important factor for FAME production from canola oil via the transesterification reaction based on the reactivity trend of SiO2@ZrO2-CaO In order to prove the critical role of the CaO and m-ZrO2 as the authors claimed, SiO2@ZrO2 catalysts without CaO should be tested for control experiments. What is the y-axis of Figure 11? How FAME production (%) was defined? Is it conversion or yield? The authors need to be more specific to report data. The authors claimed the CaO and m-ZrO2 contributions for FAME production, but I think FAME production rate just linearly increases with zirconium (VI) butoxide concentration (from 0.04 to 0.08M). As shown in Figure 11, the FAME production (I am not sure what this means as I said before) over SiO2@ZrO2 0.04, 0.06, and 0.08 M-CaO were nearly 42, 63, and 82%, respectively. This data potentially suggests that the increase in FAME production is merely associated with Zr loading, not the important role of CaO and m-ZrO2 as the authors claimed because there was no abrupt reactivity increase when Zr phases were changed to m-ZrO2 from c-ZrO2.

Reviewer 2 Report

The paper by D. Sanilas et al. deals with Ca doping into the ZrO2/SiO2 catalyst for transesterification. The introduction provides sufficient background and includes most of the relevant references. The experimental part is well described and allows potential researchers to repeat all analyses. Results are clearly presented and conclusions are supported in the obtained results. In general, I found this paper well written and I recommend it to be published as is the Journal Materials (ISSN 1996-1944). Small grammatical errors and typos should be corrected on the proof level.

Reviewer 3 Report

The paper focus on The effect of the ZrO2 loading in SiO2@ZrO2-CaO catalysts for the production of FAME from canola oil transesterification.  The author investigated also catalytic cracterisation using  TEM. XRD, BET.  In general, the results mostly support the authors' conclusions. Moreover, the originality, mechanism, and scientific reliability of the work are clear. In my opinion, there are some minor points that the authors should address before it is accepted for publication.

1°) title : trasesterification reaction --> transesterification reaction

2°) please add more refs in order to explain how the calcium oxide can improve the total basic sites

3°) what about the BET of catalyst used after 4 reuses  

Round 2

Reviewer 1 Report

I think the authors address the concerns I raised previously. The manuscript can be accepted now.